# Semi-automated socio-anthropologic analysis of the *medical discourse* on rheumatoid arthritis: Potential impact on public health

**Christine Nardini**[1]*, **Lucia Candelise**[2,3]*, **Mauro Turrini**[4]*, **Olga Addimanda**[5]

**1** Consiglio Nazionale delle Ricerche, Istituto per le Applicazioni del Calcolo "Mauro Picone", Roma, Italy, **2** ISS, Istitut Sciences Sociales, Université de Lausanne, Lausanne, Switzerland, **3** CEPED, Centre Population et Développement, Université de Paris, Paris, France, **4** Institute of Public Goods and Policies (IPP), Spanish National Research Council (CSIC), Madrid, Spain, **5** UOC Medicina Interna ad Indirizzo Reumatologico, Ospedale Maggiore, AUSL Bologna, Bologna, Italy

* christine.nardini@cnr.it (CN); lucia.candelise@unil.ch (LC); mauro.turrini@cchs.csic.es (MT)

**Data Availability Statement:** All relevant data are within the manuscript and its Supporting Information files.

## Abstract

### Background

The debilitating effects of noncommunicable diseases (NCDs) and the accompanying chronic inflammation represent a significant obstacle for the sustainability of our development, with efforts spreading worldwide to counteract the diffusion of NCDs, as per the United Nations Sustainable Development Goals (SDG 3). In fact, despite efforts of varied intensity in numerous directions (from innovations in biotechnology to lifestyle modifications), the incidence of NCDs remains pandemic. The present work wants to contribute to addressing this major concern, with a specific focus on the fragmentation of medical approaches, via an interdisciplinary analysis of the *medical discourse*, i.e. the heterogenous reporting that biomedical scientific literature uses to describe the anti-inflammatory therapeutic landscape in NCDs. The aim is to better capture the roots of this compartmentalization and the power relations existing among three segregated *pharmacological*, *experimental* and *unstandardized* biomedical approaches to ultimately empower collaboration beyond medical specialties and possibly tap into a more ample and effective reservoir of integrated therapeutic opportunities.

### Method

Using rheumatoid arthritis (RA) as an exemplar disease, twenty-eight articles were manually translated into a nine-dimensional categorical variable of medical socio-anthropological relevance, relating in particular (but not only) to *legitimacy*, *temporality* and *spatialization*. This digitalized picture (9 x 28 table) of the medical discourse was further analyzed by simple automated learning approaches to identify differences and highlight commonalities among the biomedical categories.

### Results

Interpretation of these results provides original insights, including suggestions to: empower scientific communication between *unstandardized* approaches and basic biology; promote

**Funding:** The authors received no specific funding for this work.

**Competing interests:** The authors have declared that no competing interests exist.

the repurposing of non-pharmacological therapies to enhance robustness of *experimental* approaches; and align the spatial representation of diseases and therapies in *pharmacology* to effectively embrace the *systemic* approach promoted by modern personalized and preventive medicines. We hope this original work can expand and foster interdisciplinarity among public health stakeholders, ultimately contributing to the achievement of SDG3.

## Introduction

Chronic inflammation is a known trigger and symptom accompanying noncommunicable diseases (NCDs). NCDs are responsible for 40 million deaths per year and were recognized as a silent pandemic [1] long before the dramatic rise of the COVID-19 transmissible infection. Chronic inflammation starts as a subtle alteration of the inflammatory response, whose impact remains hidden for years before the effects become overt and allow a clear diagnosis of the accompanying ailment. Therefore, the management of chronic inflammation represents an important component as well as a tremendous burden for sustainable development, as clarified by the ambitions of the United Nations Sustainable Development Goals (namely, SDG 3, [2]) and further regional [3] and national implementations [4].

Several approaches are being defined and explored to leverage the problem with biomedical research, to identify novel and more effective anti-inflammatory therapies [5], and to create health policy strategies to cope with low-cost high-impact solutions [6]. Our aim is to contribute to exploring how the management of NCDs is currently conceived and presented from another viewpoint, namely through an interdisciplinary analysis of the *medical discourse*. The *medical discourse* defines how medical practitioners and researchers (or better, a core group of experts committed to research on clinical practice) scientifically appraise the best therapeutic options available for a specific condition [7]. Contrary to the biomedical understanding, and somewhat surprisingly, this approach is far from uniform or standardized. Indeed, the medical discourse provides an arena open enough to host the plurality of conceptions of a disease as well as the techniques and practices to treat it, along with the power relations among these different approaches, ultimately defining which therapies will be offered in the clinical routine.

Studies at the intersection of anthropology, sociology of science and medicine have shown how such (pre)conceptions shape both biomedical research and health policy strategies, for example in the etiological account of a genetic disease [8], the regulation of a genetic test [9], or in the usage of support tools for medical decision-making [10]. In the current work, analyses of socio-anthropological categories are devised to explore the dichotomous labeling that characterizes the current therapeutic offer (*pharmacological* vs *non-pharmacological*, *mainstream* vs *experimental*, *standardized vs unstandardized*). Given the complexity of the task and its vastness, we narrow our focus along three lines.

First, our analysis is run from the *biomedical* standpoint, i.e. we explore the medical discourse as it is presented in *biomedicine*. Biomedicine, despite its numerous flavors [11], is generally understood as "the investigation of pathological situations using biological theories and methods: the laws of the normal applied to pathological bodies" [12]. Nowadays this includes modern disciplines like bioinformatics, computational and systems biology [13] as well as network medicine [14,15], and we use this term to indicate the most extensively recognized medical approach, in other words, the *clinical gold standard*. To take this into account we use PubMed (https://pubmed.ncbi.nlm.nih.gov/) as our reference database. Second, as an exemplar ailment to drive our analysis we use rheumatoid arthritis (RA). RA is a recognized model

## OVERVIEW ON THE ANALYSIS OF THE MEDICAL DISCOURSE

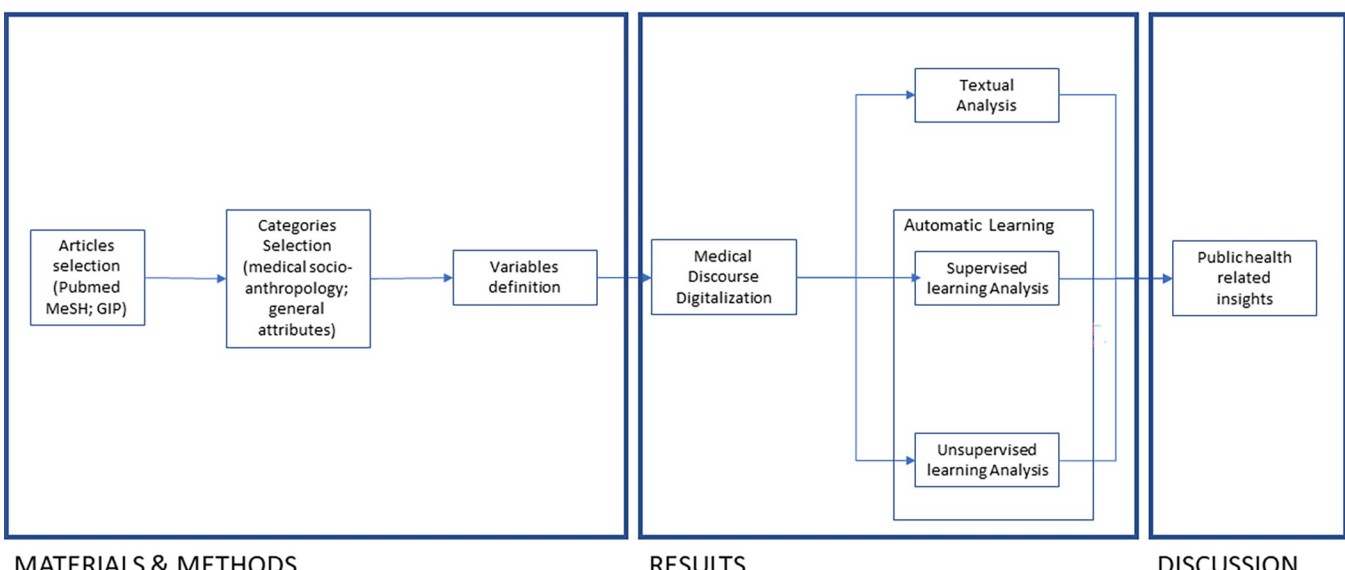

**Fig 1. Schematic representation of the phases of the analysis of the medical discourse on RA and the corresponding sections of this article.**

of pathology for chronic, inflammatory autoimmune diseases, and hence all related considerations are, in principle, translatable to other NCDs. This implies that the therapeutic gold standard, pharmacological, is defined by regional associations like the American College of Rheumatology (ACR [16]) and the European Alliance and Asia Pacific League of Associations Against Rheumatisms, EULAR [17] and APLAR [18]. Third, to identify other therapeutic options we adopt the framework represented by the *greater inflammatory pathway* (GIP, [19]), a recent conceptualization of inflammation which includes, in addition to the immune response, other types of disease-fighting mechanisms, namely: (i) the role of the gut intestinal (GI) microbiome, (ii) the role of the autonomous nervous system (ANS); (iii) the reactions to physical stimuli, described under the umbrella of *wound healing*. This represents a blueprint to build our queries on PubMed to include all RA therapies modulating the GIP functions (see Materials for details).

Our approach, as shown in Fig 1, is based on scientific reviews, a privileged source for the qualitative analysis of biomedical practices or techniques [20,21]. Our specific intention is to make these qualitative aspects explicit, viewed through the lens of a number (*nine*) of socio-anthropological categories: *typology* of the article–either *systematic reviews*, i.e. data driven statistical on multiple clinical studies, or *overviews* that collect different types of scientific publications; the *temporality* involved [22]–either characterized by a long practice—*experiential*—or *innovations*; their *legitimacy* [23,24]–either *alternative* or *complementary* in relation to other approaches; and the *spatializations* in the body [22,25] of the disease's *etiology* and *treatment* i.e. where, in the body, the ailment or therapy is thought to be operating–either *locally* or *systemically*.

In addition we collected a number of health assessment parameters [26,27], the *authors' background*; *year* of publication and information on the *geographic* localization.

Finally, manual curation of all such categorical variables, article-wise, is built into a tabular format (see Table 2) representing a global, digitalized picture of the medical discourse. This

type of representation enables the use of automated tools for investigation and, in particular, what are known as *supervised* and *unsupervised* machine learning approaches.

## Materials

### Data

Scientific reviews, based on a selection of existing research papers, are committed to carrying out a factual evaluation of a particular topic. Accordingly, they tend to hide differences and ambiguity and define, more or less implicitly, the legitimate boundaries of a specific practice or technique [20,21]. Our intention is specifically to make these aspects explicit through *nine* socio-anthropological categories (see Methods Variables).

Twenty-eight reviews were retained from a literature search of PubMed run using Medical Subject Headings (MeSH [28]) to make up a three-part query. The first part, common to all queries, broadly searches for "Arthritis, Rheumatoid/therapy"[MeSH] AND "Review" [Publication Type]". The second and third queries refine the first with specifiers including: modulators of the GIP [19] (level I) and their physical nature (level II). See Fig 2 for dependencies and S1 Table for the complete text of the queries. Level I includes the above-mentioned gut intestinal microbiome, autonomous nervous system and wound healing. Level II is taken by the list of stimuli (optic, mechanic, electric, magnetic) that are routinely used in wound healing testing [29]. The list of results obtained by all queries was narrowed down by excluding reviews that were: not on RA; not in English; not appropriate; too specific; too general, or misclassified (see Supporting Information).

The number of articles resulting from each query varied widely, ranging from three items for VNS (vagus nerve electrical stimulation of the ANS) to hundreds for pharmacological anti-inflammatory drugs. The list was finally manually curated to retain the two most recent and representative reviews for each therapeutic approach. There are some exceptions (see details in Supporting Information): (i) for the query on *anti-inflammatory drugs*, which represents the dominant biomedical approach to RA and alone offers a sufficiently long temporal excursus, i.e. an historical perspective on its evolution, we preserved eight articles; (ii) for acupuncture, in addition to the Cochrane review [30] (preferentially selected, when available), we kept two more coeval articles [31,32], building on the same material (meta-analysis of eight clinical studies, six of which were on the same subject, with both of these articles including the only two studies that are in common with the first article [30]) yet characterized by a very different *discourse* and medical conclusions; (iii) for magnetism and massage only one article could be retrieved with the above defined criteria; (iv) no article on FMT applied to RA was available, however a clinical study is ongoing (Clinicaltrial.gov, NCT03944096). Identification of the levels of the variables (see below) was inferred, where not available, in the trial record from other FMT studies [33]. For the sake of readability, from now on all 28 items in our selection are referred to as *articles*.

To further enhance the discussion we introduce three additional categories, representing therapy macro-clusters: PHA, EXP and USTD. In particular, the pharmacological gold standard therapy (from now on PHA) includes drugs aimed at the symptomatic control of inflammation, such as *non-steroidal anti-inflammatory* drugs (NSAIDs), and paracetamol or morphine(-derived) *analgesic* drugs; *corticosteroids*, which operate beyond symptoms and attempt to counteract the degeneration associated with the disease; and finally *disease modifying anti-rheumatic drugs* (DMARDs) able to interfere and block more (biologic, bDMARDs) or less (conventional, cDMARDs) specific targets, the activity of pro-inflammatory molecules (cytokines) or cells, once the disease has been diagnosed [16–18].

The approaches dealing with anomalies in the functions making up the GIP include: (i) modulators of the *GI microbiome*: diet, nutraceutics, antibiotics and fecal microbial transplant

(FMT) and (ii) modulators of the *ANS*, which coincide to date with vagus nerve stimulation (VNS). Both of the latter represent novel experimental therapies (*EXP*). Finally, we included (iii) *wound healing* modulators in the form of optic, mechanic, magnetic and electric stimulation-based therapeutic approaches, offering limited standardization, evidence-based research, guidelines, funding and overall public recognition, despite widespread use. We call the approaches falling under this third point *unstandardized non-pharmacological (USTD)*. Detailed results of the query are included in S2 Table, and we highly recommend reading the detailed description of the variables available in Supporting Information.

### Variables

As mentioned above, we extracted the *typology* of the article from each review (distinguishing *systematic reviews* i.e. data driven statistical elaboration of multiple clinical studies, and *overviews* that collect in a less systematic way different types of scientific publications, including pilot and exploratory studies); examined the *temporality* involved [22] (whether the treatments have a long practice in biomedicine—*experiential*—or are *innovations* based on promising areas of investigation); their *legitimacy* vis-à-vis the dominant pharmacological paradigm [23,24] (beign either *alternative* or *complementary* to other approaches) and the *spatializations* in the body [22,25] of the disease's *etiology* and *treatment* (*local* or *systemic*), i.e. where, in the body, the malady or the therapy are thought to be operating. In addition, we collected the *diagnostic criteria* as well as disease activity indices or health assessment parameters (ACR/EULAR relying on clinical, molecular as well as functional parameters assessed by patients and doctors, using in particular the Disease Activity Score, DAS28, to objectively assess the number of affected joints [26], OMERACT[27] or Range of Motion, ROM, i.e. the measurement of movement around a specific joint), the *authors' background* (extracted by the proxy of the main author's *affiliation*); the *year* of publication and the *geographic* localization (institute of affiliation, as a loose proxy of cultural environment). Full details are given in Methods and Table 1.

All articles were read and final values were discussed and agreed upon by all researchers.

Finally, a few words are needed to clarify the mathematical jargon needed to explain and justify the automated analysis: *variables* also represents the *dimensions* of the socio-anthropological space, and the *values* of the categorical variables are interchangeably referred to as *levels*.

The dimensionality of the nominal levels (number of values that a given variable can take) has importance in the selection of variables. When the dimensionality is high, comparable to the number of items, it becomes uninformative for automated analyses. Intuitively, each value is (almost) unique to an item; therefore generalization, groupings and patterns cannot be identified. For this reason Table 1 differentiates the low and high levels of dimensionality of the variables.

## Methods

### Digitalization

Each of the twenty-eight articles has been transformed into a vector $v_j^i, i = 1..N_A, j = 1..N_T$, with $N_A$ = 28 the number of articles selected, and $N_T$ = 9 the total number of variables. Juxtaposition of these vectors provides the digitalized representation of the medical discourse shown in Table 2, further explored via two main approaches: textual analysis and automated learning. Two different approaches are needed owing to the informative nature of variables characterized by high or low levels of dimensionality (see Materials Variables). When there are too many levels and they are uniformly distributed, it becomes meaningless to work with

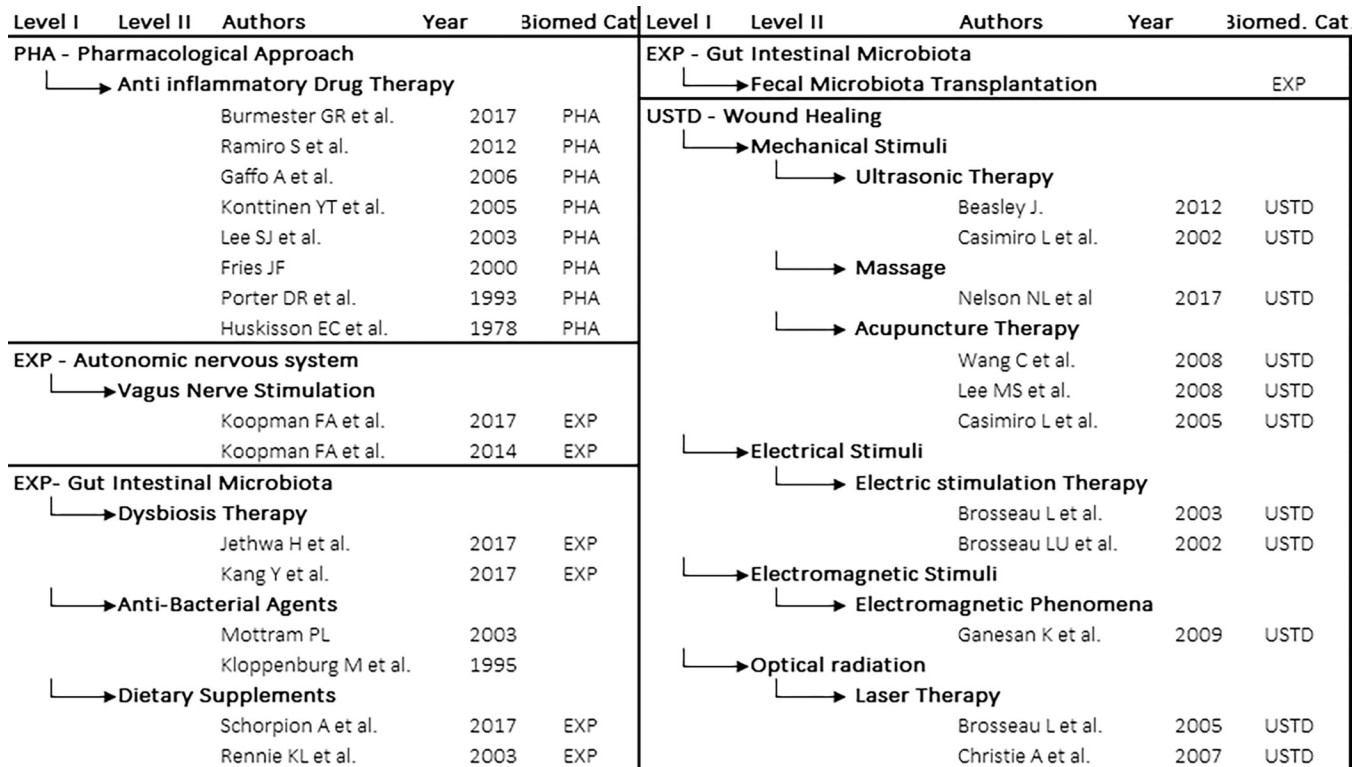

**Fig 2. List of publications (first author, date of publication) grouped by query.** Owing to space constraints only minimal information has been retained, full data is collected for review in Supplementary Materials and S1 and S2 Tables directly importing the MeSH database PubMed search. * The article does not have a doi, PMIC is used instead. * *no article was found in relation to FMT and RA, however a clinical trial is ongoing.

frequencies (absolute majority) as no variable will dominate and all values will be NA (not available). This is the case for Year of Publication, Geography and Background, which were therefore discarded from the automated analysis. However, the historical progression of PHA, the exclusive Eurocentricity of the scientific production, and the almost perfect overlap between RA background and PHA are characteristics that deserve discussion in the textual socio-anthropologic analysis.

## Textual analysis

This represents highlights and a deeper focus on the extensive work of manual curation that underlies the discretization of all articles for all variables we retained. This approach was also deemed useful as a contextualized introduction for the following automated learning analyses.

## Automated learning

For the six variables characterized by two to four levels (Typology, Temporality, Spatialization Etiology and Therapy, Criterion and Legitimacy), automated analysis was possible and was run with two further subsets of approaches. All plots and analyses were run in R (version 4.0.2) using *ggplot2* [34] and *superheat* [35] packages.

**Supervised.** To identify a *representative* vector for each of the three *biomedical categories* (*PHA*, *EXP*, *USTD*), as a supervised approach we used the simplest form of *majority voting*. Intuitively, in the representative vector, the variable's level is chosen as the one that has the

**Table 1. Variables characterizing the study.**

| Area | Variable | Nature | Values (Levels) | #Levels |
|---|---|---|---|---|
| **Socio- Anthropological** | **Temporality** | **Discretized** | *experiential, innovative, syncretic* | **Low (≤4)** |
| | **Legitimacy** | | *reference,alternative, complementary,alt-complementary* | |
| | **Spatialization Etiology** | | *local, systemic* | |
| | **Spatialization Therapy** | | *local, systemic* | |
| **General** | **Typology** | | *systematic review, overview* | |
| | **Diagnostic Criterion** | **Categorical** | *ACR/EULAR, OMERACT, ROM* | |
| | **Background** | | *MeSH term* | **High** |
| | **Geography** | | *Country code* | |
| | **Publication Date** | | *Year* | |

*List of the nine categorical variables used in this article. From left to right, the first column identifies the* Area *in which the categories useful to describe the articles' medical discourse were selected; the* Variable *column identifies the names taken or derived from the corresponding socio-anthropological categories or general attributes; the third column reports the mathematical* Nature *of the qualitative variables so defined; the Values column reports the result of the process of discretization, or the levels of the categorical variables; the column* #Levels *notes the dimensionality of the variable values (i.e. how many levels a variable can take, corresponding to the different values that can be observed in each column of Table 2), a criterion that guides the analysis presented in the Results (textual or automated).*

majority frequency (if any) across all articles in the given category. When absolute majority (frequency above half of the cases) does not exist, the value is left empty (NA).

In more formal terms, for each biomedical category $C_i$, $i = 1..N_C$, $N_c = 3$, each retained socio-anthropological variable $v_j$, $j = 1..N_v$, $N_v = 6$ and each level (value) of the categorical variable $l^i_{z_j}$, $z = 1..N_{l_j}$, with $N_{l_j}$ the number of levels in variable $v_j$, the $N_c = 3$ *reference* vectors' values $V^i_j$ are built as follows:

$$v^i_j = \left\{ \begin{array}{cc} V^i_j, & if \ V^i_j = \max_{z=1..N_{l_j}} \left( \dfrac{\sum_{i=1}^{N_{C_i}} l^i_{z_j}}{N_{C_i}} \right) > \dfrac{N_{C_i}}{2} \\ \\ NA, & V^i_j \leq \dfrac{N_{C_i}}{2} \end{array} \right\}$$

**Unsupervised.** To search for *emergent properties*, given the low dimensionality of the space and the dataset, statistical methods were not applicable. Therefore, we chose a graphical approach to visualize up to six dimensions (variables). Intuitively, we positioned the articles–represented as points–in a six-dimensional space and looked for areas where these points are denser. That is, patterns were searched for by visual inspection of the clusters, identified as areas where items are closer to each other than they are to other points in the space. In particular, the items are dots represented by the variables' graphical distributions on a scatter plot, mapping the six variables of interest to the x and y axes (Typology and Temporality), using facets to break up the two Spatializations (Etiology and Therapy) and having the points' shapes and colors to map Criterion and Legitimacy, respectively.

## Results

In this section we present the results deriving from each of the methodologies described above (digitalization, textual analysis and automated learning, and within this, supervised and unsupervised analyses), maintaining in the text of the article the same titles to highlight the matching between Methods and Results.

## Digitalization

The manually curated translation of each of the twenty-eight articles into a nine-dimensional categorical variables vector, i.e. the digitalized picture of the medical discourse, is shown in Table 2.

Its analysis proceeds in the three steps shown in Fig 1. First, a discussion on individual variables is offered (textual analysis) in regard to the *Date* of publication, *Background* of the authors and *Geography* (see Methods). Second, the automated processing (via simple machine learning) of the six other variables as a whole (system) allows different types of *emergent*

**Table 2. Digitalized summary of the medical discourse.**

| Cat. | Label | Authors | Typology | Temporality | Sp. Etiology | Sp. Therapy | Legitimacy | Criterion | Date | Background | Geo |
|---|---|---|---|---|---|---|---|---|---|---|---|
| PHA | AI | Burmester GR et al. | sys. rev. | innovation | localized | localized | reference | ACR/EULAR | 2017 | Rheumatology | DE |
| PHA | AI | Ramiro S et al. | sys. rev. | innovation | localized | localized | reference | ACR/EULAR | 2012 | Rheumatology | PT |
| PHA | AI | Gaffo A et al. | sys. rev. | innovation | localized | localized | reference | ACR/EULAR | 2006 | Rheumatology | US |
| PHA | AI | Konttinen YT et al. | sys. rev. | innovation | localized | localized | reference | ACR/EULAR | 2005 | Medicine | FI |
| PHA | AI | Lee SJ et al. | sys. rev. | innovation | localized | systemic | reference | ACR/EULAR | 2003 | Rheumatology | US |
| PHA | AI | Fries JF | sys. rev. | innovation | localized | systemic | reference | NA | 2000 | Rheumatology | US |
| PHA | AI | Porter DR et al. | overview | experiential | localized | systemic | reference | NA | 1993 | Rheumatology | UK |
| PHA | AI | Huskisson EC et al. | overview | experiential | localized | systemic | reference | NA | 1978 | Rheumatology | UK |
| EXP | VNS | Koopman FA et al. | overview | innovation | systemic | localized | alternative | ACR/EULAR | 2017 | Rheumatology | NL |
| EXP | VNS | Koopman FA et al. | overview | innovation | systemic | localized | alternative | ACR/EULAR | 2014 | Rheumatology | NL |
| EXP | Dys | Jethwa H et al. | overview | innovation | systemic | systemic | complementary | NA | 2017 | Medicine | UK |
| EXP | Dys | Kang Y et al. | overview | innovation | systemic | systemic | complementary | NA | 2017 | Inf. Dis. Medicine | CN |
| EXP | AB | Mottram PL | overview | experiential | localized | localized | complementary | NA | 2003 | Medicine | AU |
| EXP | AB | Kloppenburg et al. | overview | experiential | localized | localized | complementary | NA | 1995 | Rheumatology | NL |
| EXP | Diet | Schorpion A et al. | overview | innovation | systemic | systemic | complementary | ACR/EULAR | 2017 | Rheumatology | US |
| EXP | Diet | Rennie KL et al. | overview | innovation | systemic | systemic | complementary | NA | 2003 | Nutritional Sciences | UK |
| EXP | FMT | Zhang X. et al. | overview | innovation | systemic | localized | complementary | ACR/EULAR | 2019 | Rheumatology | CN |
| USTD | US | Beasley J. | sys. rev. | experiential | localized | localized | complementary | NA | 2012 | Occupational Med. | US |
| USTD | US | Casimiro L et al. | sys. rev. | experiential | localized | localized | complementary | ACR OMERACT | 2002 | Phys.& Rehab. Med. | CA |
| USTD | Mass | Nelson NL et al | sys. rev. | experiential | systemic | systemic | alternative | ROM | 2017 | Applied Kinesiology | US |
| USTD | AP | Lee MS et al. | overview | experiential | systemic | systemic | alt-complementary | ACR/EULAR | 2008 | Oriental Medicine | KR |
| USTD | AP | Casimiro L et al. | sys. rev. | experiential | systemic | systemic | complementary | OMERACT | 2005 | Phys.& Rehab. Med. | CA |
| USTD | AP | Wang C et al. | overview | experiential | systemic | systemic | alt-complementary | ACR/EULAR | 2008 | Rheumatology | US |
| USTD | EL | Brosseau L et al. | sys. rev. | experiential | localized | localized | complementary | OMERACT | 2003 | Phys.& Rehab. Med. | CA |
| USTD | EL | Pelland L et al. | sys. rev. | experiential | localized | localized | alternative | NA | 2002 | Physiotherapy | CA |
| USTD | LLT | Brosseau L et al. | sys. rev. | experiential | localized | localized | complementary | OMERACT | 2005 | Phys.& Rehab. Med. | CA |
| USTD | LLT | Christie A et al. | sys. rev. | experiential | systemic | localized | complementary | OMERACT | 2007 | Rheumatology | NO |
| USTD | EM | Ganesan K et al. | overview | syncretic | systemic | systemic | alternative | NA | 2009 | Biotechnology | IN |

*An editable version is available as S3 Table. Cat. = Biomedical category, Label gives the name of the therapy where AI = anti-inflammatory drug, VNS = vagal nerve stimulation, Dys = therapy working on dysbiosis; AB = antibiotics; FMT = fecal microbiota transplantation; US = ultrasound; Mass = massage; AP = Acupuncture; EL = electrostimulation including TENS and electroacupuncture; LLT = low laser therapy; EM = electro-magnetism.*

properties to be identified, defining the biomedical categories from a novel socio-anthropological perspective and gaining original insight into their power relations (via supervised and unsupervised analysis).

In fact, the two approaches enable two different types of queries. The supervised analysis tells us how the socio-anthropological variables describe three categories we care about i.e. PHA, EXP and USTD. In other words, this analysis lets us describe PHA, EXP and USTD in terms of the six socio-anthropological variables. Conversely, the unsupervised analysis notes how the levels of the six variables group together spontaneously in our data, i.e. the six variables can be used not only to describe the three known categories PHA, EXP, USTD, but also new, yet unnamed characteristics, which are discussed quadrant by quadrant.

### Textual analysis–individual variables

**Background.** Authors' backgrounds play different roles in the presentation of treatments. At one end of the spectrum are authors who are pivotal in actively supporting (innovative) therapies, as is the case for VNS (whose authors are pioneers in the approach, *overviews* mostly) or nutrition (where nutritionists explain how this medical specialty should pay attention to patients who are interested in having a suitable diet). At the opposite end (generally for therapies that are *experiential*, as well as for pharmacological treatments), we have *systematic reviews*, in most cases edited by the Cochrane Musculoskeletal group in the 2000s (between 2002 and 2005) and written by the same authors (three authors appear in all five articles we selected). It is particularly interesting to note that among these authors only one appears to be working in a rheumatology department, and none of the other authors has affiliations directly involved in the treatments examined. In these cases, therefore, the *systematic overview* collects and compares statistical data from clinical studies evaluated on the basis of formal criteria and without any practical knowledge of the dynamics associated with the treatment, except mainstream therapies.

**Geography.** The geographic origin and knowledge of a language also affect the selection of the data used to construct the articles. Namely, the article on drug therapies published in 2005 [36] by a group of researchers from Finland repeatedly points out that the approach presented is specific to northern European countries; the Cochrane Review on acupuncture [30] includes only articles in English and French, despite the fact that a large proportion of such studies are carried out in China and South-East Asia and are therefore (also) published in Asian languages. Finally, the geographical distribution of the reviews in Table 2 clearly posits the possibility of a Eurocentric bias of the biomedical discourse.

**Publication date.** This variable is particularly interesting for the PHA category, as articles cover four decades of research, showing how the dominance of pharmacology has radically evolved. The two earliest articles [37,38], published before 2000, stand out as *overviews* and recommend combining anti-inflammatory drugs (non-steroidal analgesic) with other medicaments, notably painkillers (paracetamol) or immunosuppressors (methotrexate, hydroxychloroquine, sulfasalazine as second line drugs to control progression of the disease) as well as other initiatives such as splintage, intra-articular injections, physiotherapy, hydrotherapy, occupational therapy, aids and appliances, psychological management and correct footwear [37], and a multidisciplinary approach with input from orthopedic surgeons, occupational therapists, physiotherapists, nurses, orthotics and prosthetics departments and chiropodists [38]. Interestingly, at that point in time the authors remark that, despite the hope placed in immunosuppressive and combination treatments, there is no clinical evidence proving that evolution of the disease has been halted. The articles suggest new biological treatments as a prospect for the future.

The four articles [36,39–41] published between 2000 and 2006 are *systematic reviews* and generally much more detailed in the description of RA as a systemic inflammatory autoimmune disease. Despite the richness and complexity of the text, no words are expended to describe the relationship with other approaches: no other type of treatment is mentioned beyond pharmaceutical therapy, which becomes the only reference. Consistent with the hopes raised in the two earlier articles, the focus is on new therapies and strategies (*sawtooth* [39]), with particular emphasis on the prospects the new biological drugs appear to offer, especially if started at early stages of the disease.

The last two articles [42,43] are also *systematic reviews*, and describe the most recent pharmacological approaches in terms of molecule combinations and the aims of the therapy (avoiding recurrence, improving quality of life and achieving *remission* [43]) as the only viable options. No doubts are raised about medical legitimacy, and while non-pharmacological therapies are mentioned, they are not discussed.

This concise excursus shows how articles on anti-inflammatory drugs stand out as the dominant approach: not only, in fact, does the available biomedical literature by far exceed in number other approaches but, also, these articles tend to present the therapy as the dominant and, hence, *conventional* one for RA. So much so, that by the 2000s the very meaning of *remission* appears to be altered in clinical practice. In fact, while "remission is defined as either the reduction or disappearance of the signs and symptoms of a disease" (Wikipedia: remission (medicine)), the very low rate of actual remission (20–25% [44]), moved the acceptable end points of the therapy to "low grade disease activity," creating a fluid area where RA *remission* is used to indicate stabilization of the disease, i.e. the halting of its progression [41], "remission-like state" [44], not its reduction below the threshold of activity (DAS28 ≤2.2).

The dominant position of PHA also emerges in the other articles in our selection that frame PHA more or less explicitly as the reference. The pharmacological approach is often presented in the opening section as the unavoidable reference treatment. Also, with few exceptions (acupuncture [31] and massage [45]), clinical trials present pharmacological control cohorts, despite highlighting limitations and (severe) adverse effects, that are often the argument to present the existence of a multiplicity of therapeutic approaches [46,47].

Other articles do not aim to present the impact of a single complementary therapy, but tend to oversee a plurality of approaches [47,48]. In such cases, they even go so far as to discuss the impossibility of an actual *remission*.

The year of publication also gives another insight into these therapies: some of them seem to be outdated for RA, e.g. antibiotics, while electrostimulation and laser show very early publications with no follow-up, mirroring (for antibiotics) discontinued research due to no success or possibly indicating a lack of interest and/or of funding.

## Automated learning—Systemic approach

Although the above analysis focuses on one variable at a time, it almost always highlights recurrences or influences from other variables. In order to extract additional relationships that may not emerge directly from manual curation, machine learning approaches are applied for a systemic analysis of Table 2 from which Date, Geography and Background have been removed (see Methods and S1 Fig in S1 File).

With this reduced representation we attempt to address two major issues. The first is: how socio-anthropological variables can help to frame the biomedical categories (i.e. what keeps biomedical categories distinct in a socio-anthropological framework, see the Section *supervised analysis*). The second, complementary, issue is: how socio-anthropological variables can help identify common ground among biomedical categories (i.e. what makes biomedical categories similar in a socio-anthropological perspective, see the Section *unsupervised analysis*).

## Reference Biomedical Categories

| | Temporality | Legitimacy | Typology | Criterion | Sp.Etiology | Sp.Therapy |
|---|---|---|---|---|---|---|
| **USTD** | experiential | | sys. rev. | | systemic | localized |
| **EXP** | innovation | complementary | overview | | systemic | localized |
| **PHA** | innovation | reference | sys. rev. | ACR/EULAR | localized | |

**Fig 3. Vectorial representation of the medical discourse referring to the three biomedical UnSTanDardized, EXPerimental and PHArmacological categories of approaches using socio-anthropological categories.** Each row of the above table represents a biomedical category in the form of a six-dimensional vector, where each dimension contains the representative socio-anthropological value for the corresponding category. The representative value is obtained as the majority frequency, across all representations within a category. Colors are only used to distinguish elements, with no other meaning; their values are directly labeled in the cells of the table. Cells in gray represent NA values, i.e. variables for which there is no majority; in other words, there is no dominant value for this variable.

### Supervised analysis—Socio-anthropological biomedical categories

We were interested in assessing whether there exists a common (or *reference*) socio-anthropological representation of each biomedical category. An intuitive approach to solve this matter is given by a simple *majority voting* algorithm, i.e. given all the articles belonging to one biomedical category, we identify its representative vector as the one having, for each of the six variables, the value that represents the absolute majority (i.e. the three rows of Fig 3, see Methods). As shown in Fig 3, visual, column-wise exploration makes it possible to easily grasp the unique and shared characteristics of each category.

**PHA** is uniquely *Legitimized* as the *reference*, dominated by the *ACR/EULAR criteria* and by a *localized spatialization of the etiology*, while conceptualization on the *spatialization of the therapy* is neglected or left ambiguous (NA). While the two former results are expected, given the role of the ACR/EULAR in biomedicine, the latter may be somewhat surprising given the emphasis that state-of-the-art medicines (Preventive Predictive Personalized and Participatory Precision, 3-5P medicines [11]) give to systemic approaches capable of preventing diseases via early diagnoses with observation of variables that go beyond localized symptoms [49,50].

**EXP** is the only category dominated by the *overview Typology* of articles. This is expected and understandable due to the definition of *experimental*, which implies that there only exists a limited number of the prior studies normally needed to compile systematic overviews. This

category is also majorly defined as *complementary* (Legitimacy). This is quite interesting as despite EXP approaches being both *innovative* and (partially) *systemic*–two features in line with the ambitions of state-of-the-art/future medical approaches–this does not move EXP, in the medical discourse, from an ancillary therapeutic position as *complementary* approaches to PHA.

**USTD** is uniquely characterized by the lack of a dominant diagnostic criterion and by an *experiential Temporality*. The former is interesting as USTDs do not exclude standards but simply seem to freely choose from among the available options, disregarding the dominance of the ACR/EULAR. The latter reflects two aspects that enhance each other: on the one hand, the lack of recent USTD reviews (see S1 Table) makes it difficult to assess whether more recent works might have touched upon the latest findings in basic biology (the inflammatory reflex or the relationship between mechanotransduction and inflammation [51–53]), and at the same time highlights the discontinued scientific production of reviews on USTD. It is not easy to assess whether this is due to the lack of funding offered by biomedicine dominated by PHA, or to an inertia from within the USTDs to modify the perception of Temporality, but this might be worth further exploration.

Shared values are also relevant, and interestingly our approach allows a direct comparison to be made between EXP and USTD, which in reality never cross-reference nor mention each other, i.e. in the body of the articles.

**EXP-PHA** share *innovation*, thus possibly representing the future of medicine, with the former complementing the latter.

**USTD-PHA** share the *Typology* dominated by *systematic reviews*. This is, however, representative of two very different situations, better explained by the analysis done on Background: in PHA the authorship is dominated by rheumatologists. This likely contributes to the unity of views presented in this category, and also identifies well the closed circle of experts, clearly overlapping with the rheumatology specialty. RA, however, is defined as an autoimmune musculoskeletal disorder, so possible integration of other expertise may still fit within the biomedical paradigm while opening up a broader conception of the ailment. USTD authorship, being dominated by non-rheumatologists and mostly relating techniques the authors do not apply themselves, directly affects the scientific narrative and its conclusions.

**EXP-USTD** share the conception of the spatialization of the disease characterized by a *systemic etiology* and a *localized therapy*, somewhat highlighting the hiatus between the theorization of the progression of medicine and its clinical practice, where systemic conceptualizations are more easily found in therapeutic approaches that are outside of the dominant paradigm (PHA). While the date of publication must be taken into account to correct for this trend, we can observe that the recent articles in our selection (from 2017 on) still confirm this observation.

## Unsupervised analysis—Emergent properties from the medical discourse

Given the overall small sample (28) and space (6) size, we chose the scatter plot represented in Fig 4 to position the articles in a six-dimensional space and let visual clusters emerge that represent groups of articles sharing coherent socio-anthropological coding, beyond the boundaries of their biomedical categorization. In the following, a description of the groups observed in Fig 4 is offered quadrant by quadrant, counterclockwise.

**Quadrant I** collects therapies that share a *localized etiology* and a *systemic therapy* conception, unique to PHA. In the earliest reviews in our selection ([37–40] see Table 2) RA is indeed considered a localized disease, i.e. focusing on joints and cartilage, whose treatment, however, is considered to be systemic.

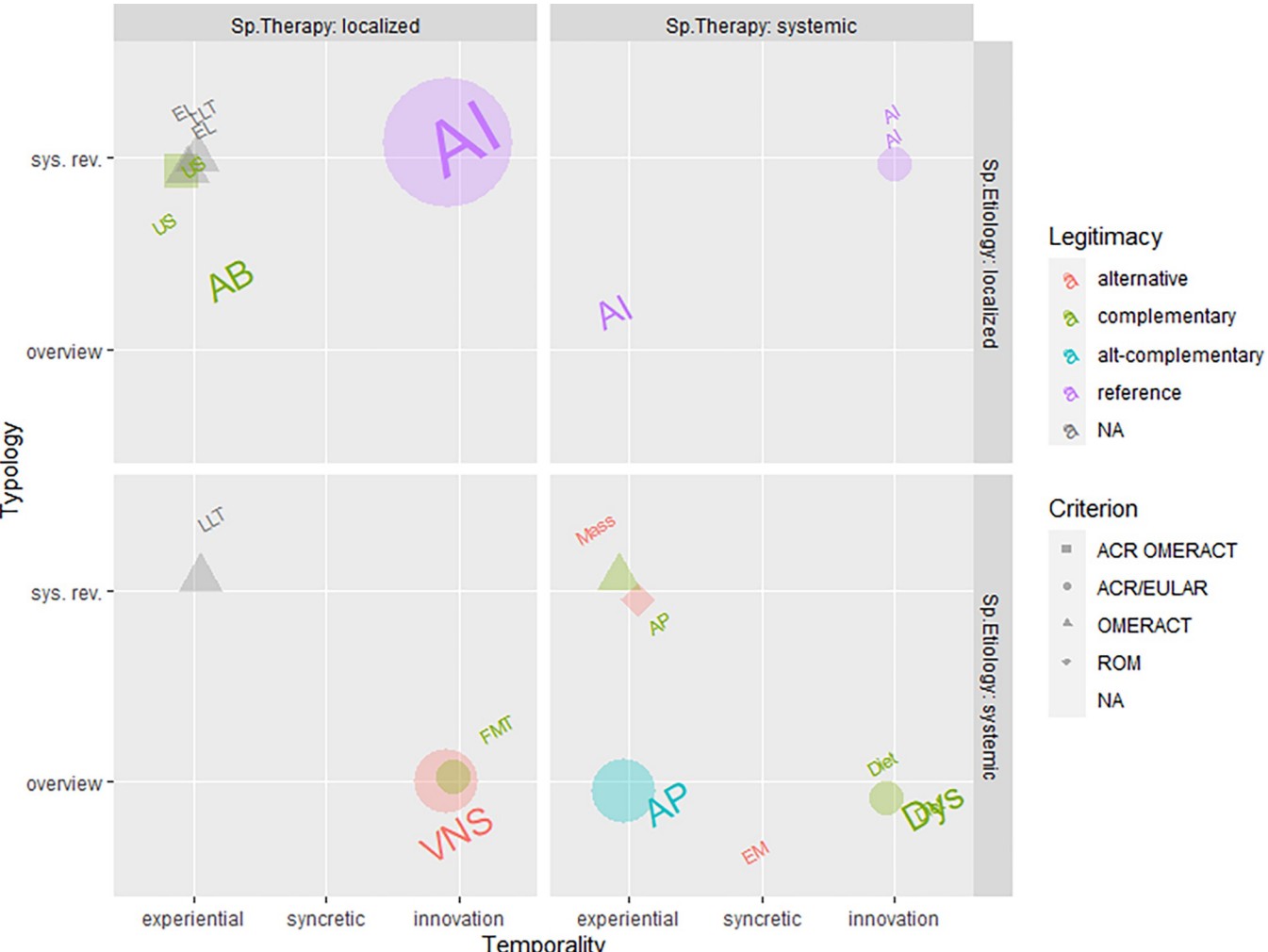

**Fig 4. Graphical representation of six of the variables under study.** Each point, whose size is proportional to the number of occurrences in our selected articles and whose color refers to the Criterion adopted, represents a therapy characterized. Labels on the points are taken from Table 2. Starting from the top right side (Quadrant I) counterclockwise to the bottom right side (Quadrant IV) several groups (clusters) are described, representing emergent properties of the medical discourse.

**Quadrant II** contains therapies characterized by a coherent (i.e. both for etiology and therapy) *local* spatialization. This spatialization is characteristic of the most recent PHA [36,41–43], which evolves its perception of the etiology from systemic (Quadrant I) to localized, and by the group of *complementary* USTDs (US, EL, LLT). *Localized etiology* for PHA means translating the pathological mechanism into the biochemical interactions among molecules, cells and tissues, whereas for USTD it means locating the disease where the body hurts.

Quadrant II is clearly broken (left from right) by Temporality as well as Criterion (green-gray versus purple) and Legitimacy (all shapes versus circles). While the right side includes only Anti-Inflammatory (AIs) drugs, on the left side are therapies that root their strength in *experience* and freely refer to a variety of diagnostics or to no standards (OMERACT, ACR/ EULAR in combination or none, all shapes of green/gray dots). These characteristics are also accompanied by being ancillary (*complementary* Legitimacy) approaches to PHA. This category includes not only USTD, as already observed in the supervised analysis, but also the obsolete antibiotic approach (AB).

In particular, the clustering approach allows us to observe that USTDs refer in this Quadrant II solely to the use of electrical and optical stimuli, completely distinct from mechanical and electromagnetic stimuli, all in quadrant IV, i.e. with an opposite spatialization. EXP therapies never fall in Quadrant II. That is to say, as mentioned already, the most innovative therapies are not localized.

**Quadrant III** is poorly populated: systemic etiology *and* localized therapy are rare medical representations. They include, however, the cutting-edge approaches of VNS and FMT, the only EXP legitimating themselves as *alternative*. This offers another viewpoint on the barrier that separates categories, already appearing with the analysis of the Background (rheumatology vs others), and indicating in opposing spatializations (local vs systemic) an additional category for segregation.

Finally, **Quadrant IV**, densely populated, can also be subdivided into two clusters: one collecting all mechanically (and EM) stimulated USTDs, and one collecting *complementary* EXP. Here PHA never appears, i.e. it is never associated with a coherently systemic representation of the ailment. The same observation made in Quadrant II returns, regarding the distinction of USTDs by localization (quadrants II and IV) with types of stimuli. While magneto- [54,55], mechano- [56,57], electro- [58,59] and opto- [60,61] sensing are recognized abilities of all our cells, only mechanotransduction appears to be framed in a coherently localized spatialization and presents itself as *alternative*, while electrical and optical stimuli require a coherently systemic spatialization, with diversified *legitimacy*.

## Discussion

From the point of view of the textual analysis, strong heterogeneity of the analyzed literature was observed in regard to contents, form and authors. This heterogeneity carries with it hierarchical relationships. They concern not only the rather predictable centrality of the gold standard treatment (PHA), but also a whole series of mechanisms that generate asymmetries. In general, approaches closer to biomedicine are depicted as more promising than experiential unstandardized approaches. When analyzed from a socio-anthropological perspective, a clear dichotomy appears between articles written by teams directly involved or generally interested in a particular therapy and articles written by authors who are not specialists in the field. The interest in the background of the authors, therefore, showed that it was possible to apply the anthropological categories of *etic*, meaning a point of view *external* to the culture studied (here the therapy reviewed), and *emic*, indicating a standpoint internal to the culture under study [62]. Interestingly, an **article using an *emic* or *etic* perspective can affect the way a treatment is framed, and generally overlaps with the concept of *overview* (emic) and *systematic review* (etic)**. This distinction allowed us to observe from another perspective the articles (*systematic reviews*) concerning numerous USTD, in most cases edited by the Cochrane Musculoskeletal group, whose members' background is only in one case explicitly affiliated to a rheumatology department. These articles are statistical meta-analyses considered to be more objective being based exclusively on clinical trials, the standard approach to produce evidence in contemporary medicine. Yet subtle biases can be traced in these reports. In fact, randomized, double blind placebo controlled clinical trials are the most statistically challenging and are therefore considered the gold standard [48]. However, the Harmonized Tripartite Guide [63] (which standardizes worldwide the statistical approaches behind clinical trials) offers a plurality of study designs. This plurality is necessary to accommodate not only non-pharmacological treatments that, for example, barely deal with blinding, but also degenerative conditions, for which placebo controls are not ethical (non-inferiority clinical trials are in this case a viable, alternative model design). Still, a design without blinding is sometimes erroneously translated into a

quality ranking and/or associated to lack of standardization, which represents a different and independent issue. As a consequence, we observe that meta-analyses are run discarding what are, in fact, appropriately designed non-inferiority trials, biasing the meta-analysis conclusions, as is the case when comparing the articles on acupuncture [34–36]. In the case of acupuncture, moreover, two out of three reviews show a clear Eurocentric bias in the choice of articles written in English, despite the fact that this technique originated in Asia and is much more widespread there.

With respect to the automated learning analysis, interesting patterns have emerged. *Spatialization*, for instance, seems a useful concept to navigate the relationships existing among the biomedical categories.

As currently emerging from our analysis, USTDs share a coherent spatialization (be it *local* or *systemic*), suggesting that these approaches refer to a unified representation of the mechanism of disease and healing. This is explicitly stated in practices defining medical action as a means to remove the obstacles that lead to the disease (consider energy stagnation in Traditional Chinese Medicine). This represents a different conceptualization from PHA, which operates downstream of processes gone awry (along the inflammatory cascade for instance), with less emphasis on the causes of the disease. Interestingly, this observation also emerges in the results of anthropological research on complementary and alternative medicines (CAM). Several studies show that in the discourse of conventional physicians practicing CAM, the global or holistic approach (which chooses to deal with the causes of the pathologies more than with the symptoms) is one of the central reasons for their choice of therapeutic approach [64,65].

Currently, state-of-the-art conceptualizations of biomedicine (3-5P medicines [11,66]) focus on the importance of systemic approaches, which implies a *systemic* spatialization of both the *etiology* (going beyond the genetic and integrating environmental factors and their molecular surrogates) and the *therapy*, with a particular emphasis on lifestyle and nutrition. Nevertheless, in our analysis, a coherent *systemic spatialization* is promoted by ancillary (complementary) nutritional therapies (EXP) and mechano-based non-pharmacological approaches (USTD). Conversely, state-of-the-art therapies (recent AI and *alternative* EXP) promote a localized vision of the treatment. Further, given the *legitimacy* they refer to (*reference* or *alternative*), one approach (VNS) seems to attempt to replace the other (PHA), with the same spatialization. This work might then recommend that the transition to systemic/ holistic approaches promoted by future 3-5P medicines are likely to require explicit attention, discussion and revisitation of the concepts that frame the perceived spatialization of the etiology of a disease.

It is also worth focusing on the distinct spatialization that USTD mechanosensing (and magnetosensing) imply versus electric and optic signals. Mechanosensing and mechanotransduction have long been known to impact tissues beyond the single cell, given the continuity with the extracellular matrix [67,68], naturally implying the necessity for a systemic approach. This is ignored in the AP reviews but mentioned in the review on massage [69], thus explaining the apparently surprising *legitimacy* of this article (*alternative*). Possibly, the growing availability of 3D-bioprinting may clarify the relevance of the environment and hence also of the systemic approach for stimuli of a different nature [70], thus offering the scientific bases for translation into clinical practice. Overall, however, the basic biological knowledge on the transduction of physical stimuli fails to be taken into account in USTDs, a fact that requires a change in the paradigm that addresses therapeutic stimuli. This may occur by explicit health policy decisions, or may be implicitly driven by technology. Just like biology and medicine renovated themselves, triggered by the technological progress of high-throughput biology [13], slowly integrating informatics to design the current network medicine paradigm [15], similarly

integration of physics and engineering pulled along by 3D bioprinting may give birth to a more complete biophysical medicine.

Strictly replicating these observations are the ones deriving from *Temporality*, where EXP and USTD are represented very differently. The former are optimistically described as promising techniques that are based on a wealth of findings from current research and aimed towards the future. In contrast, the latter are systematically described as being based solely on experience with very limited attempts to integrate biological research and experiential practices. Acupuncture provides a very clear illustration of this situation: while meta-analyses are generally characterized by purely statistical considerations, with no reference to the therapeutic nor biological rationale, we observe in [30] an attempt to forcefully refer to the biological mechanisms underlying the therapy using "*electroatropines,*" thus propagating a concept presented once by Dumoulin [71] in the 1980s with no other relationship to modern experimental biology.

This consideration calls for a two-way revisitation of the perception of USTD: from outside and from within. In fact, while the example above clearly illustrates a bias against the reliability of USTDs, it is intriguing to observe from a manual search (which did not arise from our automated queries) that a similar and opposite approach can be found in more recent literature. As may have already been seen, VNS and electroacupuncture share a very clear common ground: electrical stimulation of the vagus nerve. It is therefore interesting to observe that the most recent (2018–2021) reviews on RA and AP consist of: a meta-meta-analysis (data is extracted from prior meta-analyses, [72]) conducted in South-America, reporting the overall lack of impact of such therapy on RA, and two meta-analyses conducted in Asia [73,74] describing, conversely, a positive impact from the treatment. Only the article by Shang et al. [73] was an accessible open source and, to the best of our knowledge, a rigorously designed study. Still, the limited space for discussion given to the biological mechanism includes two research articles published in national or specialized literature (Acupuncture Research [75,76]), disregarding knowledge on the inflammatory reflex [77,78] and on other associated biological findings [51,79]. This is even more intriguing as a number of scholars of Chinese origin promote in high-impact journals on basic research the connection between the most recent findings in neurophysiology and acupuncture [80].

Clearly, in this case, contrary to the recommendation to integrate multiple Backgrounds to enhance integration from a Spatialization viewpoint, Geography, or better the medical background, could be the focus for integration to overcome the Temporality dichotomy, for example, a multicentric East-West study of acupuncture might lead to innovative conclusions.

*Legitimacy* offers yet another standpoint. We defined *alternative* as a relative concept and were indeed able to identify therapies presenting themselves as potentially self-contained, distinct from the PHA gold standard. This is not the case for *complementary*, as therapies turned out to always be complementary to the reference PHA and never to the other two classes. To date, therefore, *complementary* therapies are in fact ancillary to PHA in a clear hierarchical relationship. There is thus room to explore balanced complementarity between EXP, USTD and PHA, likely playing with timing, i.e. exploiting (the prevalence of) these different biomedical categories at different stages of the disease (prevention, early/acute stages, chronicity). This prioritization may benefit from being guided by cost-effectiveness, exploring the availability of non-pharmacological modalities of USTD and EXP to maximize safety, dissemination and ease of use. This approach may open up a third suggestion, i.e. development of the novel area of *therapy* (versus drug) *repurposing*, done by exploiting the numerous existing biomedically approved devices devised to deliver USTDs, in novel clinical studies whose design should be guided by the biological mechanisms at the roots of physical transduction and EXP therapies, in potentially extremely cost-effective ways. As already mentioned, the interaction between VNS and electroacupuncture is indeed only embryonic, knowing for instance that auricular

acupuncture stimulates a branch of the vagus nerve in the only anatomical location where it emerges on the surface of the body [81]. A recent publication offers a case for overcoming this compartmentalization [82]. Finally, not foreign to these issues is the distribution of the *diagnostic criteria* adopted across biomedical categories. Possible measures to take into account in this observation include, but are not limited to, expanding the variables that contribute to the calculation of novel scores like DAS28, including qualitative variables mirroring systemic aspects of well-being, such as sleep quality, mood, and ability to heal wounds [49,50,83].

*Typology* also provides room for considerations, meta-analyses should not be constrained by Background nor Geography, and in particular it may be relevant to integrate VNS and TENS or AP studies given the right conditions (stimulation of points in proximity to the vagus nerve), and more control should be done to ensure that the full range of approved study design is appropriately included in meta-analyses, a somewhat trivial but not unnecessary recommendation.

Overall, we observe that the medical discourse is a pluralistic space able to accommodate perspectives that are very heterogeneous. At the same time, the relationships among the different approaches are hierarchical, with unstandardized therapies marginalized and pharmacological treatment being the dominant one, and this can result in a lack of dialogue. Making all this explicit could be the first step towards a novel integration.

Clearly our analysis suffers from the overall limited number of articles we used, resulting from the necessary manual curation to identify the variables of interest (expansion and criticism in the direction of higher automation of this analysis may be the object of future work [84]). Further, our attention is turned here to the perception of *non-pharmacological* therapies, due to their potential cost-effectiveness (with an eye to *frugal innovation* [85], given the numbers and costs at stake). For this reason other *pharmacological experimental* therapies are not taken into account at this stage (for instance, but not limited to, those provided by nanoparticles or stem cells [5]).

## Conclusions

We can highlight three major conclusions of this work.

### Limitations

Limitations of this work refer mainly to the sample size and the space of investigation, which can be expanded or modified. This is not only true with respect to the categories we focused on, but also with respect to the analyses we chose for this pioneering exploration. Different types of supervised analyses, and even different choices of mapping between the axes and the socio-anthropological variables, could allow different clusters to appear. We observe, however, that several of the core patterns emerge across all our analyses, letting us infer that they may be strong enough to be considered for further investigation.

### Generalization

In close continuity with the above-mentioned limitations is the generalizability and scalability of this approach. In principle the choice of RA is purely instrumental to the background of our research partnership, therefore generalization (also including the fact that RA is a *model* disease) has no theoretical obstacles. Scalability is certainly a more relevant aspect: the manual curation of this body of articles has been, unsurprisingly, extremely time consuming. Machine learning and artificial intelligence represent an important opening in this direction, in particular *structural topic modelling* approaches, whose recent evolution [86] allows for discovery of subtle information, making it potentially suitable for analyses of the medical discourse.

## Application

How to use these types of analyses, and if and how they are meaningful and practical, cannot be decided at this point. Certainly, the compartmentalization of medical specialties has a clear reflection on the scientific discourse, impinging on conclusions owing to input data bias. The observation resulting from our analysis, that overcoming compartmentalization may require a revision of the spatial perception of the disease, is much less evident and, therefore, a valuable outcome of our work.

Our suggestions are only starting points, but we hope that their dissemination within circles of state-of-the-art medicine (EPMA www.epmanet.eu; the H2020 and HE ICPerMed initiatives www.icpermed.eu/, the Network Medicine Global Alliance www.network-medicine.org, to name a few) may trigger other richer analyses, whose results may provide tangible support to offer cost-effective means to protect citizens from the ever growing burden of NCDs.

## Supporting information

**S1 Table.**
(XLSX)

**S2 Table.**
(XLSX)

**S3 Table.**
(XLSX)

**S1 File.**
(DOCX)

## Author Contributions

**Conceptualization:** Christine Nardini.

**Data curation:** Christine Nardini, Lucia Candelise, Mauro Turrini.

**Methodology:** Christine Nardini, Lucia Candelise, Mauro Turrini.

**Writing – original draft:** Christine Nardini, Lucia Candelise, Mauro Turrini.

**Writing – review & editing:** Olga Addimanda.

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
