## [Decision Letter · Decision Letter 0]

26 Jul 2022

PONE-D-22-08328Semi-automated socio-anthropologic analysis of the medical discourse on Rheumatoid arthritis: potential impact in public healthPLOS ONE

Dear Dr. Nardini,

Thank you for submitting your manuscript to PLOS ONE. After careful consideration, we feel that it has merit but does not fully meet PLOS ONE’s publication criteria as it currently stands. Therefore, we invite you to submit a revised version of the manuscript that addresses the points raised during the review process.

 We have received comments from two independent reviewers (please see below) and they have raised some important methodological issues. Please carefully assess these comments and make the necessary changes to your submission. 

We look forward to receiving your revised manuscript.

Kind regards,

Carla Pegoraro

Division Editor

PLOS ONE

Journal Requirements:

CN was partially funded by the iPC project that has received funding from the European Union’s Horizon 2020 research and innovation programme under grant agreement No 826121. The funders had no role in study design, data collection and analysis, decision to publish, or preparation of the manuscript.

Reviewers' comments:

Reviewer's Responses to Questions

**Comments to the Author**

1. Is the manuscript technically sound, and do the data support the conclusions?

Reviewer #1: Yes

Reviewer #2: Partly

2. Has the statistical analysis been performed appropriately and rigorously? 

Reviewer #1: Yes

Reviewer #2: No

3. Have the authors made all data underlying the findings in their manuscript fully available?

Reviewer #1: Yes

Reviewer #2: No

4. Is the manuscript presented in an intelligible fashion and written in standard English?

Reviewer #1: Yes

Reviewer #2: No

5. Review Comments to the Author

Reviewer #1: The authors have presented their study related to semi-automated socio-anthropologic analysis of the medical discourse on Rheumatoid arthritis. The goal of the study is quite interesting and the results of the presented well. Some of the following concerns in the manuscript, if addressed could make the manuscript more consistent and further strengthen the author's conclusions:

1. Technically, the abstract could be slightly revised and made more precise for the users to understand the overall study.

2. Introduction explains more insights related to the study but requires few revisions and changes. For Example sentences such as “biology and biochemistry to medical research or practice” [12], including by now bioinformatics, computational and systems biology [13] as well as network medicine [14,15], …” needs revision.

3. Minor changes such as terms “data base” needs to be changed as “database”.

4. Careful revision of manuscript is required for easy understanding of the readers.

5. In the methods section, kindly modify the following sub-headings:

“Materials – reviews, Materials – Variables, Methods Analysis.”

6. In methods section, describe the unsupervised learning technique used in this study. Techniques used in this study such as unsupervised clustering is not well explained in the manuscript. Only one sentence explains it, in the discussion part.

7. This study has certain limitations such as only 28 articles are used in the analysis both for textual and automated analysis. Can this approach be implemented for other studies with a large dataset (articles)?

Search terms used for the article selection from PubMed can also be included in the methods section.

8. Careful editing of the sentences such as

“modulators of the ANS, which coincides to date with vagus nerve stimulation (VNS), which jointly represent novel experimental therapies (EXP)”

“In the following: variables represents also the dimensions of the socio-anthropological space, ..”

will make the manuscript more readable for the users.

Reviewer #2: The authors have presented a view interesting work on medical discourse related to rheumatoid arthritis. Their study is an interdisciplinary biomedical research work that need expertise in anthropology, sociology, and medicine.

Major revisions

1. Introduction section is very long. Specifically, the authors have presented a detailed note on the three stages or parts of their work in the introduction section. Instead, they can provide a summary that will motivate the readers to look into the methods immediately.

2. The authors have provided the limitations of their work in the introduction itself. This has to be moved to discussion, probably with a sub-heading, “Limitations”.

3. The definition for each variable is helpful. However, it could be short. A detailed description on each variable in the methods section may take the readers away from the main research information.

4. The authors use only six variables for the automated learning (page 12). Why? A brief note in he manuscript will be useful.

5. The automated learning section need to be elaborated for better understanding of the method implemented by the authors. How did the authors implement the automated learning? Which package did they use?

6. The results section must include the results or findings. Rest can go to the discussion section. The authors should try to present the results in Tables or Figures, not just the text (see the sub-heading, “Textual analysis – individual variables). The result section must be revised completely.

7. The authors must revise the sub-headings in such a way that the sub-heading provides or summarizes the content in the paragraphs within it.

8. Why the authors performed supervised learning on some variables and unsupervised learning on other variables? What is the reason? Can they apply either one? What happens if the learning approaches are swapped?

Minor revisions

1. “(level I) the modulators of the GIP [19],” should be “the modulators of the GIP [19] (level I),”

2. The sub-heading “Materials – Variables” should be within “Materials and Methods”. The font size must be similar to “Materials – Review”.

3. The sub-headings with Materials and Methods (e.g. Materials – Variables, Materials - Review) could be renamed for better clarity.

4. In table 1, the variables under Low and High levels is unclear.

5. Methods Analysis could have a better name.

6. PLOS authors have the option to publish the peer review history of their article (what does this mean?). If published, this will include your full peer review and any attached files.

Reviewer #1: No

Reviewer #2: No

---

## [Author Response · Author response to Decision Letter 0]

5 Aug 2022

We addressed all Reviewers' comments in the Letter uploaded, 

we verified the style and made it compliant

we added in the cover letter the modification of the funding: this study has NO funding

we added the captions for supporting information before the Bibliography

---

## [Decision Letter · Decision Letter 1]

14 Sep 2022

PONE-D-22-08328R1Semi-automated socio-anthropologic analysis of the medical discourse on Rheumatoid arthritis: potential impact in public healthPLOS ONE

Dear Dr. Nardini,

Thank you for submitting your manuscript to PLOS ONE. After careful consideration, we feel that it has merit but does not fully meet PLOS ONE’s publication criteria as it currently stands. Therefore, we invite you to submit a revised version of the manuscript that addresses the points raised during the review process.

The reviewers have identified a number of outstanding concerns that will need to be addressed before the manuscript can be accepted for publication. Please respond carefully to all of the points they have raised when preparing your revision.

We look forward to receiving your revised manuscript.

Kind regards,

Jamie Males

Editorial Office

PLOS ONE

Reviewers' comments:

Reviewer's Responses to Questions

**Comments to the Author**

1. If the authors have adequately addressed your comments raised in a previous round of review and you feel that this manuscript is now acceptable for publication, you may indicate that here to bypass the “Comments to the Author” section, enter your conflict of interest statement in the “Confidential to Editor” section, and submit your "Accept" recommendation.

Reviewer #1: All comments have been addressed

Reviewer #2: (No Response)

2. Is the manuscript technically sound, and do the data support the conclusions?

Reviewer #1: Yes

Reviewer #2: Partly

3. Has the statistical analysis been performed appropriately and rigorously? 

Reviewer #1: Yes

Reviewer #2: No

4. Have the authors made all data underlying the findings in their manuscript fully available?

Reviewer #1: Yes

Reviewer #2: Yes

5. Is the manuscript presented in an intelligible fashion and written in standard English?

Reviewer #1: Yes

Reviewer #2: No

6. Review Comments to the Author

Reviewer #1: Overall, the content and language has been improved by the authors with several changes, but there are still issues with syntactical constructions and formal scientific writing (e.g. in the added/extended text sections). Modification of subtitles and its content is more helpful for the readers to gain better insight of your research. If the manuscript is finally accepted, it should be looked at carefully for language check and sentence simplifications.

Some of the minor corrections are as follows:

1. Abstract has Background followed by Aim. In my opinion, abstract does not require the ‘aim’ section. Usually background followed by methods, results and conclusion would make it.

2. Some sentences needs simplification for readers to understand, (For example: “Our aim with this study is to contribute from yet another viewpoint, i.e. by analyzing how NCDs’ management is currently conceived and presented, using as proxy to gain insight into this issue an interdisciplinary analysis of the medical discourse.”)

3. Is ‘automated’ and ‘automatic’ learning the same in the manuscript. Kindly maintain the consistency throughout the manuscript.

4. Does this sentence explain the methodology used or the manuscript?

“Therefore, we chose a graphical approach to visualize up to six dimensions (variables) on the bidimensional space of the manuscript”. Could this sentence be modified for more clarity.

5. Conclusion section is quite long. If this could be precise, it would be appreciated.

Reviewer #2: 1. The paragraph that starts as “First, our analysis is run from the biomedical standpoint,” in page 49 and the paragraph that starts as “An overview of our approach is shown in Fig 1.” In page 50 include text on the author’s approach. The content is too long as well as clarity is missing. In the introduction, the authors should try to highlight the advancements they made. Their approach or pipeline should be presented as a summary. It must be clear and easy to understand.

2. Fig 1 can be cited in the beginning of methods. An overview of the approach (if required) can be presented withing the methods section.

3. Is the subheading “Variables” come under “Materials”? The font size shows that they both are independent subheadings. The authors should check the instructions on the subheadings allowed in the journal.

4. The authors should recheck the language in the entire manuscript. For example, “Unsupervised Second, to search for emergent properties” is not grammatically correct.

5. The authors can’t give the same subheading names within Methods and Results.

6. The authors can present the results in a better way. Current version includes only text.

7. PLOS authors have the option to publish the peer review history of their article (what does this mean?). If published, this will include your full peer review and any attached files.

Reviewer #1: No

Reviewer #2: No

---

## [Decision Letter · Decision Letter 2]

17 Oct 2022

PONE-D-22-08328R2Semi-automatic socio-anthropologic analysis of the medical discourse on rheumatoid arthritis: potential impact in public healthPLOS ONE

Dear Dr. Nardini,

Thank you for submitting your manuscript to PLOS ONE. After careful consideration, we feel that it has satisfied our scientific requirements for publication.

However, our editorial team have significant concerns about the grammar, usage, and overall readability of the manuscript. PLOS ONE requires that published manuscripts use language which is 'clear, correct, and unambiguous', see our criteria for publication at https://journals.plos.org/plosone/s/criteria-for-publication#loc-5. We therefore request that you revise the text to fix the grammatical errors and improve the overall readability of the text.

We suggest you have a fluent English-language speaker thoroughly copyedit your manuscript for language usage, spelling, and grammar. If you do not know anyone who can do this, you may wish to consider employing a professional scientific editing service.

Whilst you may use any professional scientific editing service of your choice, PLOS has partnered with both American Journal Experts (AJE) and Editage to provide discounted services to PLOS authors. Both organizations have experience helping authors meet PLOS guidelines and can provide language editing, translation, manuscript formatting, and figure formatting to ensure your manuscript meets our submission guidelines. To take advantage of our partnership with AJE, visit the AJE website (https://www.aje.com/go/plos/) for a 15% discount off AJE services. To take advantage of our partnership with Editage, visit the Editage website (www.editage.com) and enter referral code PLOSEDIT for a 15% discount off Editage services. If the PLOS editorial team finds any language issues in text that either AJE or Editage has edited, the service provider will re-edit the text for free.

Please note that we will not be able to proceed with publication of your manuscript until the concerns above are addressed.

The name of the colleague or the details of the professional service that edited your manuscriptA copy of your manuscript showing your changes by either highlighting them or using track changes (uploaded as a supporting information file)A clean copy of the edited manuscript (uploaded as the new manuscript file)

We look forward to receiving your revised manuscript.

Kind regards,

Jamie Males

Editorial Office

PLOS ONE

Journal Requirements:

Reviewers' comments:

Reviewer's Responses to Questions

**Comments to the Author**

1. If the authors have adequately addressed your comments raised in a previous round of review and you feel that this manuscript is now acceptable for publication, you may indicate that here to bypass the “Comments to the Author” section, enter your conflict of interest statement in the “Confidential to Editor” section, and submit your "Accept" recommendation.

Reviewer #1: All comments have been addressed

Reviewer #2: All comments have been addressed

2. Is the manuscript technically sound, and do the data support the conclusions?

Reviewer #1: Yes

Reviewer #2: Partly

3. Has the statistical analysis been performed appropriately and rigorously? 

Reviewer #1: Yes

Reviewer #2: No

4. Have the authors made all data underlying the findings in their manuscript fully available?

Reviewer #1: Yes

Reviewer #2: Yes

5. Is the manuscript presented in an intelligible fashion and written in standard English?

Reviewer #1: Yes

Reviewer #2: No

6. Review Comments to the Author

Reviewer #1: (No Response)

Reviewer #2: Major revisions:

1. The authors have improved the language much better than the last version. However, they need to pay more attention to correct the language. For example, “Our aim with this study is to contribute to explore” could be “We aim to contribute” or “We aim to explore”. Many sentences are too wordy and can be simplified further. This comment is applicable to the entire manuscript.

2. Introduction has improved a lot when compared to the previous version. However, the authors introduce their approach in three paragraphs. The authors should condense this to one paragraph and focus on the significant contributions they made in this current work.

3. The authors highlight the merging of manual curation into a table in the last paragraph of the introduction. It is not necessary to cite Table 2 here. Table 2 should be pushed to methods or results section.

4. In general, the introduction section does not include table or figure. Sometimes, it is okay to include a figure or a table in the introduction. However, citing a figure and two tables in the introduction is not appropriate.

5. The authors still use the same sub heading (ex. Digitalization) in methods and results. In methods, they explain their approach. In results, they present their findings. The sub heading should match this. It can’t be the same in both the sections.

Minor revisions:

1. The authors have changed the phrase to “semi-automatic” in the title. However, the phrase, “semi-automated” is the right one.

2. The authors should avoid very long sentences. For example, the following sentence is too long: “In particular, the pharmacological gold standard therapy (from now on PHA) includes drugs that aim at the symptomatic control of inflammation like non-steroideal anti-inflammatory drugs (NSAIDS), and paracetamol or morphine(-derived) analgesic drugs; corticosteroid which operate beyond symptoms and attempt to contrast the degeneration associated with the disease; and finally disease modifying anti-rheumatic drugs (DMARDs) able to interfere and block on more (biologic, bDMARDs) or less (conventional, cDMARDs) specific targets, the activity of pro-inflammatory molecules (cytokines) or cells, once the disease has been diagnosed [16–18].” The authors should try to split the content in multiple sentences such that it will be clear to the readers.

7. PLOS authors have the option to publish the peer review history of their article (what does this mean?). If published, this will include your full peer review and any attached files.

Reviewer #1: No

Reviewer #2: No

---

## [Author Response · Author response to Decision Letter 2]

8 Nov 2022

Please find below the contact name of the professional service we used to improve the readability of our work:

Lori Gerson 

Translation and Language Services

DNI: 60041731R – Calle de la Palma, 40, 3º-centro, 28004 Madrid (Spain) – tel. 639 269 818

---

## [Editor Report · Decision Letter 3]

12 Dec 2022

Semi-automated socio-anthropologic analysis of the medical discourse on rheumatoid arthritis: potential impact on public health

PONE-D-22-08328R3

Dear Dr. Nardini,

We’re pleased to inform you that your manuscript has been judged scientifically suitable for publication and will be formally accepted for publication once it meets all outstanding technical requirements.

Kind regards,

Dario Ummarino, PhD

Staff Editor

PLOS ONE

---

## [Editor Report · Acceptance letter]

16 Dec 2022

PONE-D-22-08328R3 

Semi-automated socio-anthropologic analysis of the *medical discourse* on rheumatoid arthritis: potential impact on public health 

Dear Dr. Nardini:

I'm pleased to inform you that your manuscript has been deemed suitable for publication in PLOS ONE. Congratulations! Your manuscript is now with our production department. 

Kind regards, 

on behalf of

Dr Dario Ummarino, PhD 

Staff Editor

PLOS ONE